# *Leptospira* Infection in African Green Monkeys in an Endemic Area: An Opportunity for Comparative Studies in a Natural Environment

**DOI:** 10.3390/pathogens9060474

**Published:** 2020-06-16

**Authors:** Sreekumari Rajeev, Pompei Bolfa, Kanae Shiokawa, Amy Beierschmitt, Roberta Palmour

**Affiliations:** 1School of Veterinary Medicine, Ross University, Basseterre KN 0101, Saint Kitts and Nevis; pbolfa@rossvet.edu.kn (P.B.); Kanae.Shiokawa@bio-techne.com (K.S.); ABeierschmitt@rossvet.edu.kn (A.B.); 2Behavioral Science Foundation, Basseterre KN 0101, Saint Kitts and Nevis; Roberta.palmour@mcgill.ca; 3Departments of Human Genetics and Psychiatry, Faculty of Medicine, McGill University, Montréal, QC H3A 1A1, Canada

**Keywords:** *Leptospira*, African green monkeys, Caribbean, renal lesions, zoonosis

## Abstract

This study was performed to investigate the potential asymptomatic *Leptospira* reservoir status among African green monkeys (AGMs) in the Caribbean island of Saint Kitts, and whether there is any renal pathology associated with *Leptospira* exposure. Forty-eight percent of AGMs tested were positive for *Leptospira* antibodies by the microscopic agglutination test. *Leptospira* DNA was detected in 4% of kidney samples tested using a *lipl*32 gene based PCR. We observed minimal to severe microscopic renal lesions in 85% of the AGM kidneys evaluated. The majority of the AGMs (*n* = 26) had only minimal to mild interstitial nephritis and a few (*n* = 3) had moderate to severe lesions. The presence of interstitial nephritis was not significantly associated with *Leptospira* exposure. The presence of infected AGMs in a small surface limited geographic region may pose zoonotic threat to humans and animals. The impact of *Leptospira* infection in renal pathology in AGMs warrants further investigation. AGMs residing in a natural setting in an insular, surface limited *Leptospira* endemic geographic region may offer opportunities for comparative studies to advance the field of leptospirosis. Due to their similarity to humans, such studies in AGMs may also provide translational opportunities to advance *Leptospira* research.

## 1. Introduction

Leptospirosis is a widespread zoonotic disease of global importance and may cause life-threatening illness in humans and animals. It is a prime example of the “one health” paradigm, as the interface between humans, animals, and the environment plays a major role in the maintenance and transmission of the causative bacteria *Leptospira*. Leptospirosis can be caused by numerous pathogenic serovars of spirochete bacteria, *Leptospira,* that are maintained in the renal tubules of animals and the environment [1,2]. In animals, in addition to clinical disease, *Leptospira* infection results in asymptomatic renal colonization leading to urinary shedding and subsequent contamination of the environment allowing the transmission to susceptible hosts. Various reports confirm the susceptibility of nonhuman primates to leptospirosis. Fulminant leptospirosis was reported in a captive squirrel monkey colony in French Guiana presenting with acute illness, jaundice, hemorrhagic syndrome, and mortality [3]. An epizootic event was reported in a number of captive capuchin monkeys housed in a Columbia wildlife rehabilitation center, suggesting the potential for new reservoirs and transmission routes for *Leptospira*, threatening conservation efforts and public health [4]. In the outbreak reported, jaundice and pulmonary hemorrhage were the major manifestations resulting in 71% morbidity and 27% mortality among infected animals. *Leptospira* seroprevalence has been reported in other nonhuman primates in various geographic regions [5,6]. In experimental infections, severe clinical signs similar to those seen in humans have been reported in nonhuman primates. Experimental infection in grivet monkeys has resulted in severe lymphocytic meningoencephalitis and inflammation of kidneys [7,8]. Patterns comparable to those seen in the severe forms of human leptospirosis, including pulmonary hemorrhage and severe tubulointerstitial nephritis, were present in experimental infections in marmoset monkeys [9]. Recently, we have documented *Leptospira* seroprevalence in captive and wild African green monkey (*Chlorocebus sabeus)* population inhabiting the Caribbean island of Saint Kitts, suggesting exposure to various *Leptospira* serovars [10]. In an in-depth systematic review published in 2015, a high estimated *Leptospira* related human morbidity (50.68) and mortality (2.9) per 100,000 population was reported in the Caribbean region, ranking third among WHO GBD(Global Burden of Disease) regions [11]. In a country level estimate, Saint Kitts has an estimated *Leptospira*-related human morbidity and mortality rate of 132. 71 and 7, respectively [11]. Proximity to the *Leptospira* infected animals can be a potential risk factor for other animals and humans. Our studies have also documented *Leptospira* infection and exposure in other animal species on the island [12,13,14,15,16]. In this pilot study, we investigated whether the African green monkeys (AGMs) could potentially serve as an asymptomatic source of infection and whether there is any association between *Leptospira* infection or exposure and the presence of renal pathology in a set of AGMs.

## 2. Results

This study was conducted at Saint Kitts, a small Caribbean island located in the Lesser Antilles. The AGMs (*n* = 83) sampled included 49 adults, 8 young adults, and 10 juveniles. Out of these, 50 were female, 23 were male, and sex of 10 was unavailable. All the AGMs were apparantly healthy at the time of sample collection. We tested a total of 83 AGM serum samples by microscopic agglutination test (MAT). From this cohort, 50 AGM kidney samples were available for PCR and 34 were available for histopathology examination. A list of serovars used in the MAT panel and the results are shown in the Table 1. All the samples with 50% agglutination were recorded as positive for the presence of antibodies in the screening assay at a final dilution of 1:50 and MAT titers were determined for all the positive samples.

Thirty nine out of 83 (47%) serum samples tested by MAT were positive for aggluntinating antibodies to one or more *Leptospira* serovars. MAT test was positive for 12 serovars from the MAT panel and the MAT titers ranged from 50–1600. Predominant serovars included, Bataviae (15/83:18%), Mankarso (13/83:16%), Ballum (12/83:15%), and Icterohaemorrhagiae (10/83:12%). Number of samples positive for each of the tested *Leptospira* serovars in the MAT panel is shown in Figure 1.

Agglutinating antibodies to multiple serovars were observed in the same AGMs. The majority of the samples (*n* = 21)were positive for only one serovar. Few samples had MAT titers to two serovars (*n* = 9), three serovars (*n* = 6), four serovars (*n* = 2), and six serovars (*n* = 1). Table 2 lists the MAT results from 39 individual AGM samples showing positive serovars and corresponding titers.

*Leptospira* DNA was detected in 4% (2/50) kidney samples processed for PCR, targeting the lipL32 gene with a cycle threshold value of 34.16 and 36.83. One (AGM #69) of these PCR positive samples was positive by MAT. Out of these 50 kidney samples, we examined 34 formalin-fixed and hematoxylin and eosin (H&E) stained kidney samples by routine histopathology examination. Minimal to severe lymphoplasmacytic interstitial nephritis was seen in 29/34 (85%) of the AGM kidneys. The lesions were multifocal in all affected animals, mainly seen in the cortex, with various degrees of intensity: minimal (*n* = 12), mild (*n* = 14), moderate (*n* = 2), and severe (*n* = 1). In the moderate and severe cases, various degrees of tubular degeneration, loss, and regeneration were seen. In the mild cases, focal interstitial fibrosis was seen in one sample, and multifocal tubulointerstitial mineralization (von Kossa positive- confirming the presence of Ca) was seen in nine cases. In one AGM, calcium oxalate crystals were seen in the tubular epithelial cells and tubular lumens. In the kidneys from one monkey, there were radiating arrays of pale eosinophilic, acellular spicules surrounded by macrophages and multinucleated giant cells highly suggestive of medullary tophi. There was no significant association (*p* = 1) between positve MAT results and the presence of interstitial nephritis (Table 3). Figure 2 illustrates representative lesions observed in the kidneys.

## 3. Discussion

African green monkeys, also known as vervet monkeys, are a highly adaptable nonhuman primate species inhabiting sub-Saharan Africa. AGMs were introduced in the Caribbean region by the European settlers in the 17th century and are widespread in the West Indian islands of Barbados, Saint Kitts, Nevis, and Saint Martin. In Saint Kitts, the wild population has increased drastically over the years since its first introduction [17]. The captive animals are mainly used for entertaining tourists and biomedical research. The free-roaming AGM population observed in clusters in human-inhabited areas of the island have destructive habits and a negative impact on the island’s agriculture. Trapping free-roaming AGMs is a common practice among local people, and some of these wild-caught animals are used as a source of bush meat. In Saint Kitts, a *Leptospira* endemic area, the infected AGMs, and the environment contaminated with their urine may pose a significant risk factor for humans and animals in contact.

In this study, *Leptospira* renal infection was confirmed in only two of the kidney samples tested by PCR. Only one of these samples was evaluated by histopathology, but only mild multifocal interstitial nephritis was observed in this case and this animal was negative by microscopic agglutination test. This is not surprising as we have observed similar findings in other animal species where antibody response or lesions were minimal despite the presence of organisms [12,13,18]. It is roughly estimated that 50,000 AGMs inhabit Saint Kitts, a small island with a 168 km^2^ area. Although a low-level asymptomatic infection was observed in the AGM population tested, the presence of large AGM population density in a surface area-limited geographic location is concerning. At a given time point, the magnitude of contamination and the potential risk of transmission can be proportionally high, even when a low number of animals are shedding. The handling of contaminated crops has resulted in recent Leptospirosis outbreaks in agriculture workers in Europe and Australia [19,20]. A similar zoonotic threat may exist in this island, warranting biosecurity measures in controlling contamination of agriculture products with AGM urine.

Exposure to *Leptospira,* as evidenced by the presence of agglutinating antibodies, in this cohort of AGMs is in agreement with our previous study [10]. The highest exposure was documented to serovar Bataviae, and the highest MAT titer was observed for serovar Bratislava. High-level exposure to these serovars was also observed in our previous study. Exposure to members of the serogroup Icterohemorrhagiae (serovars Icterohemorrhagiae, Mankarso, Copenhageni) was found in the population tested. We presume this to be most likely due to the cross-reactivity between members of the same serogroup. Out of two PCR-positive samples one was positive for six *Leptospira* serovars with MAT titers ranging from 50–200, suggestive of a potential early active infection. We have isolated *L. interrogans* serovar Copenhageni from rats, mongooses, and a dog from the island [12,13,15]. In rats from which we isolated *L. interrogans* serovar Copenhageni, the highest exposure was documented for serovar Icterohemorrhagiae but not for the infecting serovar Copenhageni. Exposure to serovar Ballum, a commonly found serovar in Saint Kitts rats [13] and mice (unpublished data), was also observed in this group of AGMs tested. The wild and captive AGMs are in contact with the rodents, and direct or indirect transmission from these highly infected vectors is possible in AGMs. The predominant serovar Bataviae has not been isolated yet from the island. The determination of infecting *Leptospira* serogroups cannot be extrapolated from MAT response alone because of cross-reactivity and sometimes even the absence of antibody response to infecting serovars occurring in some animal species. Cross-reactivity can be observed occasionally in some clinical cases even to unrelated serovars, a phenomenon described as paradoxical reactions [21]. The aspects mentioned above challenges the interpretation of MAT results.

Regular culturing and documentation of common *Leptospira* serovars in a specific geographic location is ideal, but unfortunately, cultures are not attempted in many prevalence studies due to the time consuming, laborious, and often futile nature of this procedure. We were not successful in isolating *Leptospira* from AGM kidney samples.

Mild to severe renal lesions were present in a proportion of AGMs tested. However, we did not find any significant association between *Leptospira* exposure and interstitial nephritis, a typical renal lesion observed in *Leptospira* infection. Recently, widespread renal lesions were also reported in another invasive species, small Indian mongooses inhabiting the island. However, a low level antibody prevalence to *Leptospira* and absence of antibody response, despite colonization in some animals, were observed in this species [18]. *Leptospira* colonizes the renal tubules and may induce host inflammatory response that may favor the removal of the pathogen. The interstitial nephritis might be an effect of previous unsuccessful colonization or might also be the response to other infectious agents. In animals, immune response to infection may have dichotomous outcomes at the humoral or cellular level, and this response may be influenced at the level of host species, individual animals, and the environment. Further detailed investigations will be needed to unravel the mechanisms leading to a likely divergent antibody vs. cell-mediated immune response and the cause of the renal lesions in AGMs.

Acute leptospirosis presented with severe kidney injury, if not diagnosed and treated, may predispose infected hosts to chronic kidney disease that may gradually proceed to end-stage kidney disease. In asymptomatic infection, bacteria may initiate inflammation that might insidiously progress to chronicity. Common *Leptospira* related renal lesions include interstitial nephritis, tubular necrosis, and degeneration that might gradually turn to chronic lesions, such as interstitial fibrosis and tubular atrophy. Chronic kidney disease of unknown origin has been reported as an endemic nephropathy across the globe [22,23]. Asymptomatic *Leptospira* colonization in renal tissue is an overlooked cause of this condition. *Leptospira* is implicated as a cause of chronic renal conditions in humans in some geographic regions and as a risk factor for nephropathy in Sri Lanka and the Mesoamerican region [22,23]. It is reasonable to hypothesize that in endemic areas, in a susceptible host, *Leptospira* infection and subsequent renal colonization can initiate an inflammatory response in the kidneys that develop to chronicity and gradual renal impairment. Therefore, *Leptospira* infection, as an emerging predisposing etiology of chronic kidney disease, warrants in-depth investigation. Chronic lymphoplasmacytic interstitial nephritis, which is not uncommon in kidneys of animals, can have many possible causes, and could be primarily inflammatory or represent foci of antigen persistence. Although the histological lesions were not deemed severe enough in most of our cases to result in clinical symptoms, mild and chronic nature suggests an attack from an infectious agent and subsequent attempt to remove the agent. The renal mineralization, presence of oxalate crystals, and tophi observed in some AGM kidneys can be considered as incidental background lesions, not directly linked to *Leptospira* infection.

AGMs are widely used for biomedical research to study human diseases [24]. They share a high degree of homology with humans and hence may serve as a useful translational model for studying leptospirosis and other diseases in a natural setting. A surface-limited insular location such as Saint Kitts, where *Leptospira* infection is highly endemic in animal populations, offers an ideal opportunity for studying various aspects of leptospirosis. We propose to undertake well-designed prospective longitudinal studies to advance our understanding of the impact of *Leptospira* infection in renal pathology in AGMs. These studies may provide translational knowledge to support *Leptospira*-related chronic kidney disease in the human population. Finally, research studies in a susceptible population in its natural environment might be useful to acquire a deeper understanding of multiple aspects of *Leptospira* infection dynamics, including maintenance, transmission, and pathogenesis. Such studies will facilitate an additional opportunity to elucidate whether *Leptospira* infection has a potential role in conditions such as chronic nephropathy reported in humans in various parts of the globe. This knowledge will ultimately benefit in the development of better prevention and control strategies for *Leptospira* infection and potential health-related consequences in human and animal populations.

## 4. Materials and Methods

Since this study was initiated on a pilot basis, we used a convenience sampling strategy based on the availability of samples for testing. All the AGM samples originated from Behavioral Science Foundation, Saint Kitts. These include captive animals (*n* = 49) residing in open cages at the facility and wild animals (*n* = 34) captured from the island and housed temporarily in the facility. All the animal procedures were performed with high ethical standards adhering to IACUC approved protocol (#TSU1.23.18) of the Ross University School of Veterinary Medicine (RUSVM).

Serum samples were obtained from 83 AGMs, and kidney samples were collected from 50 euthanized AGMs. The serum samples were screened against 21 *Leptospira* serovars (shown in Table 1) belonging to 16 serogroups using the microscopic agglutination test (MAT). The MAT was conducted using protocols described previously with modifications [10,25]. Briefly, *Leptospira* serovars were grown in Ellinghausen-McCullough-Johnson-Harris (EMJH) media supplemented with *Leptospira* Enrichment (Difco^TM^ Leptospira Medium Base EMJH, and Enrichment, Becton, Dickinson and Company, Sparks, MD, USA). A 4–7-day-old culture of each of the serovars adjusted to 0.5 McFarland standard was used for testing. In a 96-well plate, 50 μL of the *Leptospira* culture was mixed with 50 μL of each of the AGM serum samples diluted to 1:25 in PBS. After 1.5 to 2 h of incubation at 29 °C, the samples were screened for the presence of agglutination reaction to each of the serovars using a darkfield microscope. All the samples with 50% agglutination were recorded as positive for the presence of antibodies. Positive controls were set up using homologous antiserum (Royal Tropical Institute, KIT Biomedical Research, Amsterdam, The Netherlands) to each of the serovars included in the panel. Initially, all the serum samples at a final dilution of 1:50 were screened for the presence of agglutinating antibodies. Endpoint titers were determined for each of the positive samples against positive serovars.

DNA was extracted from the supernatant of homogenized AGM kidney samples using DNeasy Blood & Tissue Kits (QIAGEN Scientific Inc., Germantown, MD, USA), following the manufacturer’s protocol. TaqMan probe-based Real-time PCR (RT-PCR) was performed on a Smart Cycler (Cepheid Inc., Sunnyvale, CA, USA) as described previously [26]. The *lipL*32 gene present in pathogenic members of the genus *Leptospira* was targeted. A cycle threshold value above 40 cycles was considered negative. DNA extracted from a positive *Leptospira* culture served as positive control and sterile water served as the negative control.

Both kidneys, when available, were collected and fixed in 10% neutral buffered formalin, trimmed, processed, and stained with hematoxylin and eosin (H & E). Two board-certified veterinary pathologists evaluated renal histopathology. The association between MAT results and interstitial nephritis was compared using Fisher’s exact test in GraphPadPrism (GraphPad Software 8.30, La Jolla, CA, USA). The statistical significance was set at *p* = 0.05.

## 5. Conclusions

Our study suggests that AGMs may serve as asymptomatic reservoirs, play a role in environmental contamination, and contribute to *Leptospira* transmission to humans and animals. The impact of *Leptospira* infection in renal pathology in AGMs warrants further investigation. AGMs residing in a natural setting in an insular, surface-limited, *Leptospira* endemic, geographic region may offer opportunities for comparative studies to advance the field of leptospirosis.

## Figures and Tables

**Figure 1 pathogens-09-00474-f001:**
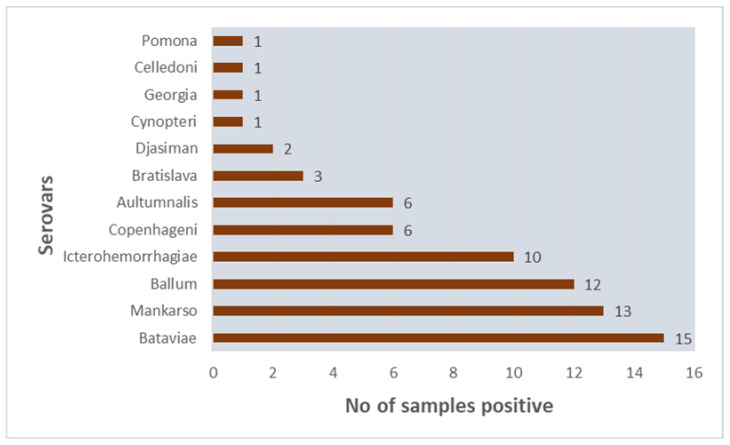
Summary of MAT results showing number of samples positive for *Leptospira* serovars.

**Figure 2 pathogens-09-00474-f002:**
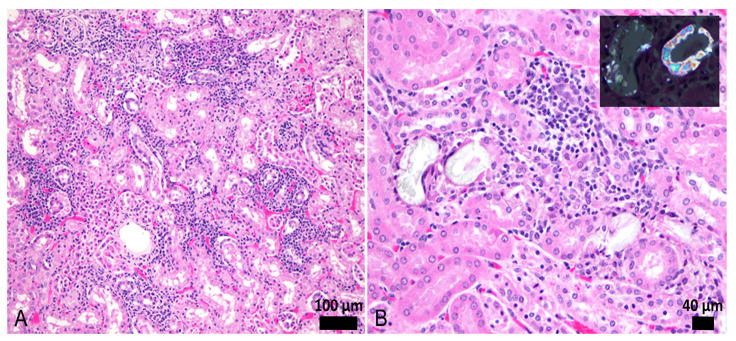
(**A**,**B**) Renal lesions in AGMs. (**A**) Multifocal lymphoplasmacytic interstitial nephritis and mild tubular epithelial degeneration. (**B**) Focal lymphoplasmacytic interstitial nephritis and the presence of intratubular calcium oxalate crystals with loss of epithelial cells and foreign body reaction with multinucleated giant cells. Inset shows crystal birefringence on polarized light.

**Table 1 pathogens-09-00474-t001:** *Leptospira* species, serogroups and serovars of strains used in the microscopic agglutination test (MAT) panel. The MAT results for each of the serovar are given in parenthesis.

Species	Serogroup	Serovar (Results)
*L. interrogans*	Australis	Australis (Negative)
		Bratislava (Positive)
	Autumnalis	Autumnalis (Positive)
	Bataviae	Bataviae (Positive)
	Canicola	Canicola (Negative)
	Djasiman	Djasiman (Positive)
	Grippotyphosa	Grippotyphosa (Negative)
	Icterohemorrhagiae	Icterohemorrhagiae (Positive)
		Mankarso (Positive)
		Copenhageni (Positive)
	Pomona	Pomona (Positive)
	Sejroe	Hardjo (Negative)
		Wolffii (Negative)
*L. borgpetersenii*	Ballum	Ballum (Positive)
	Javanica	Javanica (Negative)
	Tarrasovi	Tarrasovi (Negative)
*L. santarosai*	Pyrogenes	Alexi (Negative)
		Pyrogenes (Negative)
	Mini	Georgia (positive)
*L. kirschneri*	Cynopteri	Cynopteri (Positive)
*L. weilii*	Celledoni	Celledoni (Positive)

**Table 2 pathogens-09-00474-t002:** Distribution of positive serovars and their corresponding MAT titers (in parenthesis) in 39 individual African green monkeys (AGMs) tested positive by MAT. AGM# in the column 1 represents the individual identification number of the AGMs tested.

AGM #	Serovars Positive (MAT-Titer)
3	Autumnalis (200), Bratislava (1600)
4	Ballum (200)
5	Bataviae (100)
7	Ballum (100)
10	Ballum (400)
14	Ballum (50)
15	Ballum (50)
32	Djasiman(50)
36	Icterohemorrhagiae (200), Mankarso (50)
37	Icterohemorrhagiae (200), Mankarso(100), Copenhageni (100)
38	Bataviae (100)
39	Bataviae (100)
40	Icterohemorrhagiae (100) Mankarso(50)
41	Icterohemorrhagiae (50)
42	Bataviae (100)
43	Icterohemorrhagiae (200), Mankarso (100), Ballum (400), Bataviae (100)
45	Icterohemorrhagiae (50), Bataviae (100)
46	Icterohemorrhagiae (400), Mankarso (400), Copenhageni (200)
47	Bataviae (100)
48	Bataviae (200)
51	Bataviae (50)
53	Icterohemorrhagiae (50), Mankarso(50)
55	Bataviae (100)
56	Ballum (100)
57	Icterohemorrhagiae (400), Mankarso (800), Copenhageni (100)
66	Autumnalis (50)
67	Autumnalis (50)
69	Autumnalis (200), Cynopteri (50), Georgia (50), Icterohemorrhagiae (100), Mankarso (200), Copenhageni (100)
70	Bratislava (50)
71	Mankarso (100), Copenhageni (50)
72	Autumnalis (50), Mankarso (100)
73	Ballum (200), Bataviae (100),Celledoni (50), Mankarso (100)
74	Autumnalis (400), Bratislava (50), Pomona (50)
76	Ballum (100), Bataviae (100), Mankarso (50)
77	Bataviae (50)
78	Ballum (100)
79	Ballum (200), Bataviae (100)
81	Ballum (50) Mankarso (50), Copenhageni (50)
83	Bataviae (50), Djasiman (50)

**Table 3 pathogens-09-00474-t003:** Comparison of MAT results with the presence of interstitial nephritis using Fisher’s exact test. MAT—microscopic agglutination test; ISN—interstitial nephritis. A *p*-value less than 0.05 is considered as significant.

	ISN+	ISN−	Total
MAT+	7	1	8
MAT−	22	4	26
Total	29	5	34

(*p* = 1).

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
