# Peer review of "Leptospira Infection in African Green Monkeys in an Endemic Area: An Opportunity for Comparative Studies in a Natural Environment"

_pathogens, 2020, doi:10.3390/pathogens9060474_

Round 1

Reviewer 1 Report

Thank you for the opportunity to review your interesting manuscript. I have some suggested edits, and overall consider that more detail is needed in both the methods and the results sections.

Some specific comments are:

Introduction:

Line 51 – by morbidity rate – do you mean incidence?

Methods:

More information is needed about the population of monkeys that you have studied. Where did they come from, how were they trapped, and were the sampling methods likely to produce a representative sample?

Line 147-148: Please name serovars/ groups of the MAT panel. Also what was your cut-off for positivity?

Results:

Some comment on the age/sex and health of the monkeys is needed – especially given your focus on asymptomatic disease.

Lines 65-66 – Please be consistent with the number of decimal places used. Two is probably overly precise given the numerator and denominators.

Was there a relationship between Leptospira seropositivity and the presence of interstitial nephritis. Were all those with interstitial nephritis also seropositive/ PCR positive? If not, what other causes of interstitial nephritis did you consider and exclude? Ditto for the renal calculi

Discussion

I think lines 112-128 are of questionable relevance to the data you present. It reads (to me) as though you are indicating that your data supports the notion that asymptomatic Leptospira colonisation of humans is an important cause of ESRF (lines 120-121). As highlighted above, I think more data is needed from your study about the cause of the interstitial nephritis and renal calculi, before you can say that your study supports the view that ‘Leptospira infection and subsequent renal colonization can initiate an inflammatory response in the kidneys that develop to chronicity and gradual renal impairment’. I’m not necessarily saying that the statement is wrong, but just that your data isn’t (currently) robust enough to use as supporting evidence.

An interpretation of the reactive serogroup patterns would be useful (notwithstanding the limitations of attributing the likely infecting serogroup by patterns of reactivity).

Conclusions:

Line 159: ‘AGM may serve as asymptomatic reservoirs and may contribute to Leptospira transmission to humans and animals.’ This may be true, but please provide data that the AGMs were asymptomatic, and put some interpretation around the fact that only 2 of 50 were found to be PCR positive vs 38 of 83 that were seropositive. Might this ratio of PCR positive to seropositive indicate that for most AGM infectio

Reviewer 2 Report

General comments: This study by Rajeev et al. was performed to assess the occurrence of asymptomatic Leptospira infection among African green monkeys (AGMs) in the Caribbean island of Saint Kitts. It is known that the kidneys are the main target of pathogenic Leptospira spp. during infection of the mammalian hosts, therefore the authors also attempted to find any renal pathology associated with Leptospira exposure.

A total of 83 AGM serum samples were tested by the microscopic agglutination test (MAT, titer 50-1600) using a panel of 21 serovars as antigens. Of these examinations, app. 46% were positive. Histopathogy studies performed on 34 out of 50 kidneys revealed that the majority of the tested animals had mild interstital nephritis.

Thus, the authors concluded that asymptomatic Leptospira renal infection occurs in the AGM population in the island. Therefore the infected AGMs may generate a potential risk factor for leptospirosis in this region.

Major comments: Good scientific journals obligate and Pathogens is among them. Therefore, in my opinion, the authors should present their results much better, i.e. more graphically and more interesting. The methodology should be also descibed in more detail.

To improve this manuscript I would suggest:

1) Results of the MAT analysis shoud be presented in visually attractive way. For example, distribution of the MAT titers to serogroups (serovars) could be shown. The authors write that the MAT titers ranged from 50-1600, and the highest titer was observed for serovar Bataviae. The question therefore arises what about other serovars.

Results of the MAT analysis of the tested animals could be summarized in the table.

2) Fig. 1 - the magnification should be determined; scale bars should be also included in the figure;

3) lanes 147-148: The serum samples were screened against 21 Leptospira serovars belonging to 17 serogroups using the MAT. What were these serovars and serogroups? This information should be provided.

4) lane 19-150: Leptospira cultures were treated with serum at 1:25 dilution in PBS - What was this serum? I am sure that a panel of anti-sera was used to determine serogroups and serovars. Probably, sera were pretested at the dilution 1/25 and then these sera had to be retested to define endpoint titers. What was the minimal significant titer to assess an animal as seropositive?

Serogroup and serovar classifications by using MAT should be described in detail.

Brief description of the Leptospira cultivation should be also given in the Section Materials and Methods.

5) Real-time PCR should be also described in the Section Materials and Methods. Probably, a Tagman probe-based assay was used to detect the lipl32 gen of the pathogenic Leptospira spp. Was PCR also used in this study? If not, please, specify the methods in the Abstract and in the main text.

I wonder if the authors have tried to isolate leptospires from monkey kidneys and culture them to confirm leptospiral infection by an additional test.  Another quesion is whether urine samples were available. It happens that MAT results can give false epidemiological serovar presence in a population, therefore in this study other tests may have been used for typing and identification of Leptospira serovars, including "culture isolation". Identification of Leptospira from urine samples could reveal new serovars. Analysis of urine samples would also be helpful to assess shedding of Leptospira in AGMs in the island, and hence their contribution to pathogen transmission  to other animals and humans.

Round 2

Reviewer 1 Report

My suggested revisions have been addressed. No further comments

Author Response

Thank you very much!

Reviewer 2 Report

I appreciated the authors' work on their manuscript. However, there are still some issues that need clarification and refinement.

1) Lanes 269 - It is written that the serum samples were screened against to 21 serovars belonging to 17 serogroups and Table 1 added to the revised manuscript includes 16 serogroups. Please, enter the correct number of the Leptospira serogroups used in this study;

2) lane 70, Table 1: Leptospira species, serogroups and serovars used in the MAT panel. instead of "Leptospira species, serogroup and serovars of strains used in the MAT panel.";

3) Figure 1: Summary of MAT results showing number of samples positive for Leptospira serovars -This summary added to the revised manuscript shows that 72 serum samples were positive for Leptospira serovars. This number does not agree with the number of seropositive animals (40 out of 83 serum samples tested by MAT were positive, see lane 94). Probably, the authors have included cross-reactions between Leptospira serovars;

4) Figure 2 (added to the revised manuscript): Numbers of animals positive at a specific titer.

Also, in this case the number of animals (61) does not match the number of positive animals (40 out of 83 serum samples tested by MAT were positive, lane 94).

The authors should ensure that the reader is not misled. The cross-reactions between the tested serovars should be marked when presenting the MAT results.

5) Table 2 - Positive Serovars and number of animals positive for each of the serovar and the titers. Please, recalculate the values/number in the second and last column; the Total number of animals in the second column is 16 instead of 27; the Total number of animals in the last column is 31 instead of 72. The same comment as above.

6) lane 96: Predominant serovars included... instead of Predominant serovars incuded;

7) lane 277: All the samples with 50 % agglutination was recorded as... or All the samples with 50% agglutination were recorded as...?

Im my opinion, it would be worth adding in the Results section that serum samples with the titer of 50 (reciprocal of the final dilution of serum with 50% agglutination) were assessed positive.

8) Figure 3: The scale bars have been marked but they are not legible.

9) lane 285: "Real-time PCR (RT-PCR) was performed on a Smart Cycler (Cepheid Inc., CA, USA) as described previously [26]" and also lane 176: "In this study, Leptospira renal infection was confirmed in only two of the kidney samples tested by PCR." Could it be written that it was a TaqMan probe-based real-time PCR? In the Materials and Methods section, please, provide also information about primers used in real-time PCR assays.

The authors have added information about AGMs and therefore I would like to ask how the MAT results are distributed taking into account the sex of monkeys. What about young adults and juveniles? Are anti-Leptospira antibodies detected in these groups?
